# OpenReview forum: "CLASH: Evaluating Language Models on Judging High-Stakes Dilemmas from Multiple Perspectives"
_ICLR.cc/2026/Conference — ICLR 2026 Poster_

### Official Review · Reviewer_ycte · 2025-10-27

**Soundness:** 3
**Presentation:** 2
**Contribution:** 2
**Rating:** 4
**Confidence:** 4

**Summary:**

The paper contributes CLASH, a dataset with 345 long-form, high-stakes dilemmas in story form, along with contextualized values that may be relevant in those cases. Additionally, they evaluate understanding of decision ambivalence and psychological discomfort (a new extension compared to prior work to my knowledge). Through experiments on 14 models, they analyze differences between thinknig and non-thinking models, and also what contributes to successful reasoning traces. Additionally, they do analyses on LLM steerability.

**Strengths:**

S1: The contributed dataset is very rich and longer-form than prior work, which is an exciting contribution.
S2: The analysis of reasoning models' performance vs. the base model, along with their characteristics, is novel as far as I'm aware for value-related work.
S3: The cognitive characteristics of successful reasoning (less backtracking and verification) for moral dilemmas was very interesting.
S4: I found that the evidence that LLMs are quite sticky to their initial value preference, making them less steerable, to be an exciting noe (L407-408).
S5: I quite liked the analysis of not only how values conflict, but also predicting how people may perceive the value trade-off (e.g., ambivalence/discomfort) - very interesting direction!

**Weaknesses:**

W1: I'm not sure exactly how novel the idea of "Conditional steerability" is - I think for other prior datasets, (e.g., Value Kaleidoscope), there are also values which conflict to come to differing judgments. In the case where values do not conflict, it seems that steerability is less needed. Also, isn't all steerability conditional, as you necessarily need to condition (steer) to something? The steerability experiments are interesting - but may be better framed as merely "steerability" experiments if the distinction between steerability and conditional steerability cannot be justified. With that, I'm not sure if I buy the current description in L470-471 that "Picaco ... does not address dilemmas where values lie in direct opposition."
W2: Some of the writing involves potentially overly anthropomorphic language, e.g., "Do LLMs exhibit discomfort when making hard decisions?", when what is actually being measured is whether LLMs can detect when _humans_ might exhibit discomfort. Verbage here and throughout the paper could be tightened up to be more precise.
W3: At times, I found the presentation to be confusing. Some more concrete details about the evaluations, or potentially some examples, could strengthen legibility in an extended camera-ready version of the paper.

**Questions:**

Q1: Could the authors explain if they believe conditional steerability to be meaningfully different from prior steerability research?
Q2: The authors talk about contextualization of values being a strength of this compared to prior work. However, the prior works also contextualize values in the context of a particular situation. Could the authors clarify in what way the contextualize that goes beyond prior work, if they do so?

**Details Of Ethics Concerns:**

In the introduction, the authors note that their work focuses on assessing LLMs' ability to perform well in high-stakes environments, where "their outcomes carry significant consequences such as loss of life and substantial financial implications." (L44-45). However, nowhere do the authors engage with the idea of whether LLMs _should_ be used in these scenarios, or the potential risk of false assurance if LLMs do get high scores on a "high-stake" benchmark but fail to generalize in application.

I think that an ethics review might be best to tackle how best to consider the potentially harmful applicability of models trained for high-stakes decision-making, and whether the authors might want to at least engage with the ethical issues in the paper.

---

> ### Author Response · Authors · 2025-11-21
> **To reviewer ycte**
>
> Thank you for your comments! We present our rebuttal below:
>
> ### Conditional Steerability
> Previous work considering multiple values, including Picaco([1]) as mentioned in our paper, has tried to align models toward multiple values using multi-objective reinforcement learning ([2],[3],[4]) or by prompting ([5],[6]). However, research in moral philosophy ([7]) shows that many values are mutually incompatible in specific contexts, where agents must inevitably prioritize one value over another when no option satisfies all moral requirements. These situations are naturally formulated as dilemmas, which are common in real-life decision-making.
>
> Several works have addressed value conflicts in decision-making in dilemma situations. Value Kaleidoscope ([8]) presents the values associated with each side, but does not evaluate how an LLM can be steered towards selecting one side. DailyDilemmas([9]) examines steerability in decision-making tasks but does so by reinforcing adherence to one side rather than balancing both. Yet in genuine dilemmas, the critical aspect when making decisions is not simply which values are present but how individuals “prioritize” among them, since people often differ in which value they choose to emphasize when there is no clear-cut right decision. This motivates our introduction of “conditional steerability”, which focuses on “Can the model follow value X when it conflicts with value Y?” compared to prior work that mainly focuses on “Can the model follow value X” or “Can the model follow both value X and Y?”.
>
> We will clarify this distinction and add more references, including those presented here, in the camera-ready version of the paper.
>
> ### Verbage and Adding Details
> Thank you for pointing out the anthropomorphic wording. We agree that we should avoid anthropomorphic language, and we present revisions:
>
> L270-271: Models trade off between clear-cut and ambivalent decisions, potentially exposing limitations in model training. -> Models trade off between identifying clear-cut and ambivalent human decisions, potentially exposing limitations in model training.
> L294: Do LLMs exhibit discomfort when making hard decisions? -> Can LLMs detect the discomfort humans experience when making hard decisions?
>
> We will review the entire paper to ensure anthropomorphic expressions are removed for the camera-ready version. We will also include additional details and examples from the appendix using the extra page allowed for the camera-ready version.
>
> ### Contextualization of Values
> Yes, values were contextualized within the situations in prior work. However, our contribution lies in how this contextualization is implemented. Whereas prior works tend to make the target values explicit by directly stating them ([8], [9]), we instead embed values implicitly within the character descriptions. This design introduces an additional layer of complexity; the model must infer which values are relevant within the character descriptions and then apply those when generating an answer. Although [6] includes some value-related statements that could be viewed as implicit presentations, our method extends beyond this by introducing relationships among values within the character descriptions (different levels of prioritization, shift of values). Human value systems are not just collections of multiple values; they contain subtle interactions and trade-offs. With these relational aspects, our benchmark more accurately reflects the complexity of human perspectives. Together, this makes the benchmark more challenging and better suited for evaluating nuanced value-understanding capabilities.
>
> We also appreciate the comments regarding the ethics statement. We address your concerns by adding an **Ethics Statement** section to our paper, which we have highlighted in blue. Please refer to the updated paper.
>
> **References**
>
> [1] PICACO: Pluralistic In-Context Value Alignment of LLMs via Total Correlation Optimization, Jiang et al., arXiv, 2025
>
> [2] Beyond One-Preference-Fits-All Alignment: Multi-Objective Direct Preference Optimization, Zhou et al., Findings of ACL, 2024
>
> [3] Multi-objective Reinforcement Learning: A Tool for Pluralistic Alignment, Vamplew et al., Pluralistic Alignment Workshop @NeurIPS, 2024
>
> [4] Robust Multi-Objective Preference Alignment with Online DPO, Gupta et al., AAAI, 2025
>
> [5] SynthesizeMe! Inducing Persona-Guided Prompts for Personalized Reward Models in LLMs, Ryan et al., ACL, 2025
>
> [6] Can Language Models Reason about Individualistic Human Values and Preferences?, Jiang et al., ACL, 2025
>
> [7] The proper study of mankind: An anthology of essays, Berlin, Isaiah, 2000
>
> [8] Value Kaleidoscope: Engaging AI with Pluralistic Human Values, Rights, and Duties, Sorensen et al., AAAI, 2024
>
> [9] DailyDilemmas: Revealing Value Preferences of LLMs with Quandaries of Daily Life, Chiu et al., ICLR, 2025

---

> > ### Comment · Reviewer_ycte · 2025-11-22
> >
> > Thank you for engaging. I especially appreciate the addition of the updated ethics section!

---

> ### Author Response · Authors · 2025-11-25
>
> We would be glad to discuss further if you have any remaining concerns. If we have addressed all of them, we kindly ask you to consider raising your score. Thank you!

---

### Official Review · Reviewer_ViBw · 2025-10-28

**Soundness:** 3
**Presentation:** 3
**Contribution:** 3
**Rating:** 6
**Confidence:** 3

**Summary:**

This paper introduces CLASH (Character perspective based LLM Assessments in Situations with High stakes), a benchmark of 345 expert written, long form high stakes dilemmas paired with 3,795 perspective narratives that instantiate competing values. The dataset targets under explored aspects of value based decision making: decision ambivalence, psychological discomfort, and temporal value shifts inside perspectives. Fourteen models, both thinking and non thinking, are evaluated. The paper reports several findings: (i) even frontier systems struggle on ambivalence, with low accuracies reported for strong proprietary models; (ii) models predict discomfort better than they track value shifts; (iii) cognitive behaviors that transfer in math or games do not transfer to value reasoning, with new failure modes such as early commitment and overcommitment; (iv) conditional steerability toward a target value is negatively correlated with a model’s base preference for the competing value; and (v) third person framing often increases steerability relative to first person, with safety as a notable exception that benefits from first person framing. The authors release measurement protocols for steerability over value pairs and framing, and provide analyses across model families and sizes.

**Strengths:**

1. The dataset contains 345 expert-written, long-form dilemmas with 3,795 perspective narratives, which captures value tradeoffs, context, and temporal shifts much better than short synthetic prompts. This makes the benchmark more ecologically valid for medical, legal, and financial settings.

2. The task frames competing value pairs inside character narratives and measures decision ambivalence and psychological discomfort, two aspects that typical moral benchmarks ignore. This lets the paper probe how models reason when values conflict rather than treating values independently.

3. The paper introduces a concrete procedure that compares base preferences to steered preferences for each value in a pair, with a simple numeric mapping that yields steerability scores. This enables pairwise analysis across frameworks and supports aggregate statistics, so steerability becomes measurable and comparable rather than anecdotal.

4. Fourteen models are tested across families and sizes, covering both thinking and non-thinking modes. The analysis surfaces consistent findings, such as a negative correlation between inherent preference and steerability, and robust framing effects where third-person prompts usually help while first-person boosts Safety in particular.

5. The paper identifies early commitment and overcommitment as recurring errors in value reasoning and ties them to the evaluation setup. These findings point directly to follow-up mitigations, for example prompt or decoding controls, and give researchers concrete behaviors to test rather than only aggregate scores.

**Weaknesses:**

- “High stakes” is defined conceptually but the operational rubric and inclusion criteria are not fully specified. The perspective to value mapping and any inter annotator agreement are under reported.

- The default prompt is third-person, and the first-person version likely differs in more than pronouns. Even small lexical edits can change length, explicitness, or the presence of safety-loaded tokens. Since many frontier models apply safety policies that are sensitive to “I” or “you” phrasing, first-person prompts may disproportionately trigger refusals or policy-shaped outputs.

- The analysis spans Straightforward, Simple Contrast, and Swayed Contrast categories, which may vary in average length and in how explicitly values are named. If first-person variants are longer or contain more direct references to Safety or Self-Esteem, differences in steerability may reflect lexical salience rather than perspective per se.

- The steerability computation maps Yes/No/Ambiguous to 1/0/0.5 and then forms ratios b/a and c/d. If first-person prompts shift the base distribution closer to extremes, denominators can become small, which amplifies ratio variance and makes the first- vs third-person comparison sensitive to a few items.

- The paper filters to value pairs that occur more than 16 times, which it says is about the top 25 percent of all pairs. This immediately creates an imbalance: many value pairs are either excluded or represented by small n, while a handful dominate the statistics. Since CLASH has 345 dilemmas and 3,795 perspectives, the effective sample per specific pair can still be modest, especially once split by model family, size, or framing.

- Steerability ratios b/a and c/d can be unstable for rare pairs where base preferences cluster near 0 or 1. Without per-pair counts, confidence intervals, or multiple-comparison corrections, it is difficult to separate genuine scale effects from sampling imbalance.

- Perspectives are nested within dilemmas, so items are not independent. Ignoring this nesting inflates nominal sample size and can overstate significance.

**Questions:**

1. Can you replicate the first vs third person result with templates that only change pronouns and keep token length matched, and report invalid output rates and safety policy triggers per condition?

2. How are early commitment and overcommitment operationally detected?

---

> ### Author Response · Authors · 2025-11-21
> **To reviewer ViBw (1/2)**
>
> Thank you for finding our work valuable! We address your comments below.
>
> ### First vs Third
> Regarding the first vs third person results, we would like to clarify the following point.
> > The default prompt is third-person, and the first-person version likely differs in more than pronouns. Even small lexical edits can change length, explicitness, or the presence of safety-loaded tokens.
>
> > Can you replicate the first vs third person result with templates that only change pronouns and keep token length matched, and report invalid output rates and safety policy triggers per condition?
>
> We agree with this, so we carefully design our first-person prompt to introduce minimal differences from the third-person prompt. Please compare Figure 17: third-person prompt versus Figure 35: first-person prompt. Both prompts have a token count of 255 according to the GPT-4o token counter. We will make sure this point is clear in our camera-ready version.
> Also, there are very few invalid outputs and safety policy triggers in our dataset. Across the dataset, the percentages for first-person and third-person results are 0.03% and 0.06%, respectively. The percentages are far too small to support any meaningful conclusion. This is expected, as our dataset consists of dilemma situations in which both options are reasonable, and there is no clearly correct or incorrect answer.
>
> ### Differences of Categories
> > The analysis spans Straightforward, Simple Contrast, and Swayed Contrast categories, which may vary in average length and in how explicitly values are named
>
> We appreciate the reviewer’s concern that the different character description categories (e.g., Straightforward, Simple Contrast, Swayed Contrast, etc.) might differ systematically in narrative length or in how explicitly values are named.
>
> First, we would like to clarify that the length difference is not a confounding factor but an inherent feature of the category design. For example, descriptions in the Swayed Contrast category are necessarily longer than those in the Straightforward category. The Straightforward category is concise by definition; it describes a character who consistently prioritizes one value above all others. In contrast, the Swayed Contrast category must articulate that the character follows both values but favors one when they come into conflict, which inevitably requires additional explanation. Thus, variation in the length is a natural consequence of the conceptual distinctions between categories.
>
> To understand how explicit values are stated, we analyze the parts of each character description that are related to the underlying values. We find that these segments either match one of the supporting or opposing rationales or present a closely paraphrased version. Below, we provide an example (id: 70-scug. Please refer to the supplementary materials to access the data for this id):
>
> Action: Working on Saturday
>
> Supporting Rationale: Commitment to work and one's career
>
> Opposing Rationale: Maintaining work-life balance and spending time with family
>
> Supporting Rationale Mapped Values: Ambition, Courage
>
> Opposing Rationale Mapped Values: Friendliness, Temperance
>
> Parts that are related to values:
>
> *Straightforward:*
> - (supporting) driven by an unwavering belief in the importance of their career and professional achievements
> - (supporting) values dedication to their work and community
> - (supporting) willingly prioritizes long hours at the office and continuous professional development
>
> *Swayed Contrast:*
> - (supporting) deeply invested in their career
> - (supporting) choose professional commitments over family time
> - (supporting) drive for career advancement takes precedence
> - (opposing) value the time spent with their family
>
> None of the character descriptions, across any category, directly state the “values”, but rather implicitly convey those through paraphrased rationales. Because of this consistent implicit phrasing, the level of explicitness is effectively the same across categories.

---

> ### Author Response · Authors · 2025-11-21
> **To reviewer ViBw (2/2)**
>
> ### Steerability Experiments
> We first want to clarify some points.
> >The steerability computation maps Yes/No/Ambiguous to 1/0/0.5 and then forms ratios b/a and c/d. If first-person prompts shift the base distribution closer to extremes, denominators can become small, which amplifies ratio variance and makes the first- vs third-person comparison sensitive to a few items.
>
> The base distribution is *always* computed from the base prompt, not from the first- or third-person prompts. Thus, the denominators used for steerability computation come from a fixed baseline that does not change across settings. Because this baseline is constant, the situation the reviewer described, where first-person prompts can drive denominators toward zero, cannot occur. We update the paper to make this explicit by adding the exact base prompt used in Figure 34.
>
> > Perspectives are nested within dilemmas, so items are not independent. Ignoring this nesting inflates nominal sample size and can overstate significance.
>
> We would like to clarify that this nesting does not affect the significance analysis. We compute value preferences using the character descriptions, and then use the preferences to calculate the steerability and statistical significance. Therefore, the statistical test operates on aggregated preferences rather than raw perspective-level responses, removing the dependence on nested items.
>
> >Since CLASH has 345 dilemmas and 3,795 perspectives, the effective sample per specific pair can still be modest, especially once split by model family, size, or framing.
>
> Our intention is that steerability is a characteristic per model, so should be computed “per model”, not per model family. So, the effective sample size is not reduced because of the number of model families, size, or framing.
>
> >The paper filters to value pairs that occur more than 16 times, which it says is about the top 25 percent of all pairs. This immediately creates an imbalance: many value pairs are either excluded or represented by small n, while a handful dominate the statistics.
>
> We agree with the reviewer that the cutoff of the top 25\% of value pairs can make the results skewed. So, except for pairs with a base preference of 0 or 1, which yield a zero denominator in the computation, we include all other pairs and rerun our steerability experiment. We report the updated results as follows:
>
> The correlation between base preference and difference of steerability became 0.046 with a p-value of 3.64e-1. Qwen and Llama exhibit negative correlations of -0.044 and -0.183, with p-values of 7.05e-1 and 1.10e-1, respectively. The results indicate that when all pairs are included, these correlations are no longer statistically significant. However, when applying the original threshold (more than 16 occurrences), the correlations become significant, showing that frequently occurring pairs still reveal a statistically significant tendency.
>
> Also, we reconfirm that smaller models are significantly more likely to show greater steerability toward one value over the other (p < 0.005), while larger models exhibit significantly higher overall steerability (p < 0.0001), even when all pairs are used.
>
> We will update Section 4.3 - Findings accordingly.
>
> > How are early commitment and overcommitment operationally detected?
>
> For early commitment and overcommitment detection, we prompt GPT-4o. The prompts are included in Figure 26 - 30 for reference.
>
> Thank you for your comments!

---

### Official Review · Reviewer_Ehbp · 2025-10-30

**Soundness:** 4
**Presentation:** 4
**Contribution:** 4
**Rating:** 8
**Confidence:** 3

**Summary:**

This paper developed the new benchmark dataset named CLASH which addresses a high-stake dilemma. They also introduced the concept of ambivalence, psychological discomfort, and value shift. They found that even leading LLMs can struggle with these traits. The authors also explored Conditional Steerability. Also analyzed first and third-person perspectives.

**Strengths:**

- This paper is really easy to follow, clearly presented. Well organized.
- Originality of proposing a new high-stake dilemma dataset
- Significance may comes from introduce new concepts of measuring"Decision Ambivalence", "Psychological Discomfort", "Value Shifts"  in LLM and finding LLM having difficulties in understanding"Decision Ambivalence", "Psychological Discomfort", "Value Shifts"
- The way that how they develop their dataset is well documented.
- Valuable human annotator contributions.(With concern of IRB statement)

**Weaknesses:**

- As this paper address high-stake situations, may have potentially contain and drive the research direction to sensitive domain.
- Human annotators/inspectors are included, potentailly require IRB (Institutional Review Board) consent?
- I don't see any weakness other than this.

**Questions:**

- It is interesting to introduce the ambivalence measure, but how can we actuallly know LLM is just avoiding to choose one of the answer or recognizing the "true ambivalence"?
- Your experiments were conducted using the 'reasoning process followed by the answer' prompt strategy. Was there any pilot study or literature-based discussion on the impact of reversing this order (i.e., 'answer followed by reasoning')?

**Details Of Ethics Concerns:**

The authors describe high-stake dilemma may refer to medical, legal and financial domains (line 1) Indeed, the authors compose their dataset by: Medical (23.48%), Business (31.88%), Journalism/Media (33.33%), and Government/Politics (11.30%) (Appendix).

In my experience, even simple troy dilemma scenario investigation sometimes require jail-breaking of LLMs. If the high-stake scenario dataset and realated research is initiated, this may force researchers to jail-break. While it is grey area whether jail-breaking is violation or not, but the high-stake scenarios may include potentially sensitive topics not only restricted to those raised by the authors.

The reason why I put flag is to let community aware of this potential issues.

---

> ### Author Response · Authors · 2025-11-21
> **To reviewer Ehbp**
>
> Thank you for finding our work valuable! We present our responses below.
>
> > Human annotators/inspectors are included, potentailly require IRB (Institutional Review Board) consent?
>
> First, we would like to clarify that our study is IRB exempt. The human participants were only involved in the inspection process of the dataset for quality assurance purposes; their behavior itself was not the object of analysis.
>
> > It is interesting to introduce the ambivalence measure, but how can we actuallly know LLM is just avoiding to choose one of the answer or recognizing the "true ambivalence"?
>
> Regarding ambivalence detection, one way to assess whether the models understand ambivalence or not is by examining their reasoning chains. This allows us to determine whether the model is actually recognizing the balance between conflicting values rather than just abstaining from answering. For example, consider the reasoning chain from Claude-Sonnet-4:
>
> ---
>
> … **Character A has conflicting values here.** They want financial sustainability for quality journalism (which paywalls provide) BUT they also want equal access to information for all (which paywalls prevent). The character description emphasizes that A "strives to find solutions that respect both principles" and wants "financial health and inclusivity coexist harmoniously."
>
> Since paywalls inherently create a conflict between these two values and don't allow them to coexist harmoniously, Character A would likely find this action problematic. However, A also recognizes the need for financial sustainability. **This creates genuine ambiguity - A wouldn't find it clearly acceptable or unacceptable, but would be torn between the competing values.** …
> Answer: Ambiguous
>
> ---
>
>
> The bold parts illustrate that Claude-Sonnet-4 is actively considering the balance rather than avoiding to answer the question.
>
> >Your experiments were conducted using the 'reasoning process followed by the answer' prompt strategy. Was there any pilot study or literature-based discussion on the impact of reversing this order (i.e., 'answer followed by reasoning')?
>
> We choose the format of reasoning followed by the answer to reduce answer-induced bias in the reasoning process. Prior work shows that when models produce the answer first, the subsequent reasoning often serves to justify the answer rather than truly reasoning ([1], [2]).
>
> Finally, thank you for your suggestions for the ethics statement! We have added the ethics statement and colored it blue. Please refer to the revised version of our paper.
>
> **References**
>
> [1] Bias-Augmented Consistency Training Reduces Biased Reasoning in Chain-of-Thought, Chua et al., arXiv, 2024
>
> [2] ReCoT: Reflective Self-Correction Training for Mitigating Confirmation Bias in Large Vision-Language Models, Qu et al., ICCV, 2025

---

> > ### Comment · Reviewer_Ehbp · 2025-11-21
> >
> > I find the authors' efforts to community valuable. Hence, I retain my score.

---

### Official Review · Reviewer_8oMt · 2025-11-04

**Soundness:** 2
**Presentation:** 2
**Contribution:** 3
**Rating:** 4
**Confidence:** 4

**Summary:**

This paper introduces CLASH, a dataset of 345 high-stakes ethical dilemmas with 3,795 character-based value perspectives, designed to evaluate LLMs' ability to navigate complex value-based decisions. The scale of the benchmark they develop is significant and the 11 kinds of character perspectives are thoughtfully designed. The authors then benchmark 14 models for their abilities to detect decision ambivalence, recognize psychological discomfort, and track temporal value shifts and also examines reasoning patterns in thinking models. The paper also introduces a "conditional steerability" analysis that looks at whether models can be steered towards one value over the other using their constructed character descriptions, relative to the models' base preference.

**Strengths:**

This paper presents a genuine effort to improve resources for evaluating value conflicts in high-stakes dilemmas with a benchmark of a useful scale. The dataset's scale and structured approach to creating different perspective categories enables controlled testing of specific value reasoning capabilities.

In particular, the inclusion of dynamic value shifts (Shift, Half-Shift, False-Shift categories) is innovative and addresses an unexplored aspect of value reasoning. The findings for RQ3 were most interesting to me and seemed most relevant to real-world problems cases i.e. understanding current LLMs' ability to track evolving values in multi-turn conversations.

The authors take care to include a number of analyses beyond reporting performance on their benchmark, such as the strategies used in reasoning chains and the steerability of models' values given the character descriptions. In particular, attempting to understand the strategies used in reasoning chains in this setting is novel and interesting given that previous work has only done this for mathematical reasoning. The authors make an effort to support many of their claims with statistical tests.

**Weaknesses:**

My main concerns about this benchmark relate to data contamination and the heavy use of GPT-4o for most of the relevant content. I see that efforts were made to address data contamination issue in Appendix B.1, but testing only 10 samples and using a 60% threshold with the same model that generated the content does not seem rigorous enough for scenarios that are directly taken from text that is very likely to be in the training data. Further, the use of GPT-4o for all the actions, rationales, and character descriptions raise concerns about whether this benchmark is testing whether models align with GPT-4o's interpretation of value-based reasoning. The high performance of GPT-4o (third by a small margin) seems like a further indication of this.

The authors justify the quality of each of these generation steps with high inter-annotator agreement, but as they point out in Appendix B.4, "The high inter-annotator agreement is expected due to our use of explicit character perspectives." For example, without a perspective, a question like “Who should receive the kidney transplant: the person with higher chance of survival or the person who waited longer?” invites diverse opinions. But if we specify that Character A prioritizes fairness over utility, it is clear that A would select the person who waited longer." I think this highlights a significant limitation of the benchmark -- conflicting values in such scenarios in the real-world are diverse and that's what makes this problem challenging. Relying on one model that is well-known to be subject to mode-collapse raises concerns that benchmark does not capture realistic features of real-world value conflicts or is a good test of other model families' abilities in this domain.

In terms of the results, the claims made in section 4.2 seem unjustified given the lack of statistical tests, error bars, and the very minimal difference between the values in the graphs for these claims. For example, it is not clear from the top left subplot in Figure 4a that "successful value-laden reasoning exhibits less backward-chaining" though this might be true of verification. Similarly for "successful chains demonstrate greater emphases on pragmatic and rights-based ethics." This section would benefit from the addition of paired t-tests for these claims.

Section 4.3 is the weakest in my opinion, and could benefit from a much better justification for the complexity of mapping the rationales to such a wide space of values and then again mapping those to a different set of value frameworks. These mappings were again done primarily done using GPT-4o raising concerns about their accuracy. In Appendix F.3 the authors state that "Human evaluators indicate that 78.00% of the values are comprehensive, 98.77% are relevant to the value-related rationale, and the inter-annotator agreement measured with Cohen’s kappa score (McHugh, 2012) were 0.471 and 0.823, respectively." The fairly low assessment of comprehensiveness and the low inter-annotator agreement on this suggests that relying on GPT-4o to make complex value mappings does not form a trustable basis for the analyses in 4.3. Further, while the r=-0.243 (line 407) is significant it is still only a weak correlation.

**Questions:**

In addition to the points mentioned in the weaknesses, the paper could benefit from the following:
- The results in RQ3 seem consistent with the overcommitment and early commitment strategies the authors find in the reasoning traces in section 4.2 which would be interesting if true, however this connection isn't made in the text. Breaking down the analyses of reasoning chain strategies by the static and dynamic cases to see if these strategies are favored in the dynamic settings and connecting these back to RQ3 could strengthen the results.
- It would strengthen this benchmark to have content generated by more than one model family and evaluate models' performance across all these model families' data points.

Minor:
As the authors note in the related works, this setting is very related to moral decision-making studies that have been done in LLMs. The paper could better contextualize their contributions in light of these studies, especially more recent ones e.g. Cheung et al., 2025 (Large language models show amplified cognitive biases in moral decision-making).

---

> ### Author Response · Authors · 2025-11-21
> **To reviewer 8oMt (1/5)**
>
> We appreciate the reviewer’s careful reading and comments!
>
> ### Concerns Regarding the Use of GPT-4o and the Mapping Procedure
> > the use of GPT-4o for all the actions, rationales, and character descriptions raise concerns about whether this benchmark is testing whether models align with GPT-4o's interpretation of value-based reasoning.
>
> We emphasize that GPT-4o was used only as an initial aid for generating candidate elements of each situation. All outputs were carefully reviewed by human annotators, who validated the model-generated elements and revised them thoroughly to meet our annotation standards (please refer to the full checklists in Figure 11, 13, and 16).
>
> To further assess whether the elements are overly influenced by a particular proprietary model (gpt-4o), we also examine the similarity between our rationales and those produced by Claude-3.5-Sonnet. We select 50 situations from our dataset and prompt Claude-3.5-Sonnet to generate rationales. An independent evaluator who is not aware of our research goals compared our rationales to Claude’s and reported that 71% were similar. Note that Claude’s outputs, like those from GPT-4o, require non-trivial revision to satisfy our annotation checklist. Thus, 71% is a conservative estimate, and with comparable human refinement, the similarity would be even higher. This further supports that our pipeline is not tied to GPT-4o and that any strong proprietary model can serve as a reasonable starting point, with human oversight ensuring that the final rationales adhere to our checklist.
>
> > Section 4.3 is the weakest in my opinion, and could benefit from a much better justification for the complexity of mapping the rationales to such a wide space of values and then again mapping those to a different set of value frameworks. These mappings were again done primarily done using GPT-4o raising concerns about their accuracy.
>
> We choose to first map rationales to intermediate values and then map those to value frameworks because human-curated mappings already exist for the intermediate values ([1]). This allows us to reuse reliable annotations and avoids the substantial additional effort that would be required for humans to map our rationales directly to the frameworks. We also attempted to have language models perform the direct mapping from our rationales to the value frameworks, but the results were noticeably less reliable than mapping those to intermediate values. A likely reason is that the intermediate values (e.g., fairness, justice, transparency) are common concepts that appear frequently in text, whereas many framework-level values (e.g., secular-rational, physiological) are more abstract and less familiar to models. So we decided to use the two-step process instead. We will explain this design choice in detail in the camera-ready version.
> Also, we clarify that, for our mapping procedure (Appendix F.3), we do not solely rely on GPT-4o. Our mapping generation pipeline rather combines the results of (i) a supervised entailment model (Laurer et al., 2024) and (ii) GPT-4o as an independent entailment model. This mitigates over-reliance on GPT-4o and helps ensure that our dataset construction is robust.
>
> **References**
>
> [1] DailyDilemmas: Revealing Value Preferences of LLMs with Quandaries of Daily Life, Chiu et al., ICLR, 2025

---

> ### Author Response · Authors · 2025-11-21
> **To reviewer 8oMt (2/5)**
>
> ### Concerns regarding Data Contamination
> >I see that efforts were made to address data contamination issue in Appendix B.1, but testing only 10 samples and using a 60% threshold with the same model that generated the content does not seem rigorous enough for scenarios that are directly taken from text that is very likely to be in the training data.
>
> We appreciate the reviewer’s concerns about potential data contamination. While we acknowledge that some of the “situations” in our dataset may have appeared during training, only these situations, not the character descriptions, could have been previously seen. All character descriptions in our dataset are newly generated through our pipeline, meaning models would not have been exposed to these elements during training. Moreover, even if certain situations were seen before, it is still unclear whether existing LLMs understand the subtle dynamics involved, such as ambivalence, discomfort, or value shifts. Our dataset is designed precisely to capture these nuances and thereby fill in gaps from previous work.
>
> Our goal for the contamination experiment is to ensure that the dataset remains sufficiently challenging so that model performance on it would still be informative. We agree that using only 10 samples and using GPT-4o might not be rigorous. To address this, we conducted an additional experiment using more examples per website. Specifically, we use half of the available examples for each site (ama: 40, scub: 29, john: 26, medeng: 57, scug: 19) and evaluate performance using a different proprietary model (Claude-Sonnet-3.5). The results are as follows. Each column indicates the performance of [Character Description Category]-[Question] (e.g., Straight-G means Straightforward category - General question).
>
> |Website|Straight-G|Straight-D|Simple-G|Swayed-G|Swayed-D|Shift-P|Shift-C|Half-P|Half-C|False-P|False-C|Overall|
> |---------|-----------:|-----------:|---------:|---------:|---------:|--------:|--------:|-------:|-------:|---------:|---------:|--------:|
> |ama|98.75|84.74|35.00|87.50|89.72|98.75|96.25|100.00|28.75|98.75|75.00|81.20|
> |scub|100.00|70.69|34.48|93.10|83.33|100.00|100.00|98.28|22.41|91.38|89.66|80.30|
> |john|96.15|90.00|23.08|92.31|75.00|100.00|98.08|98.08|26.92|92.31|80.77|79.34|
> |medeng|96.49|87.19|45.61|84.21|80.85|97.37|92.98|98.25|36.84|96.49|75.44|81.07|
> |scug|100.00|84.21|21.05|97.37|92.11|100.00|100.00|100.00|26.32|94.74|86.84|82.06|
>
> Although the overall accuracy of Claude-Sonnet-3.5 is quite high, it is important to note that the Claude model family achieves the strongest results among the models we test. Even so, several categories, such as Simple-G, Half-C, remain challenging, with scores below 50%. This indicates that the websites indeed serve as a source for a sufficiently difficult and discriminative benchmark for evaluating language model performance.

---

> ### Author Response · Authors · 2025-11-21
> **To reviewer 8oMt (3/5)**
>
> ### Complexity of Real-Life Conflicts
> >For example, without a perspective, a question like “Who should receive the kidney transplant: the person with higher chance of survival or the person who waited longer?” invites diverse opinions. But if we specify that Character A prioritizes fairness over utility, it is clear that A would select the person who waited longer."
>
> >conflicting values in such scenarios in the real-world are diverse and that's what makes this problem challenging
>
> We would like to clarify our contribution: our goal in this work was to introduce a framework that enables systematic evaluation of models’ value understanding. In this context, the use of explicit character descriptions is a deliberate design choice, which is a contribution rather than a limitation. Yes, it is possible to present the dilemma directly to a large language model and obtain answers. However, these answers would simply reflect the model’s preferences, making it difficult to assess their value understanding abilities. By introducing the character descriptions, we can clearly define a ground truth answer, which transforms the task into a verifiable evaluation.
>
> We agree that our current character descriptions do not yet capture the full diversity of human values. However, we want to assert that our contribution is to provide a benchmark that can explicitly evaluate nuanced aspects of value reasoning. For future work, we plan to extend the dataset to include a broader range of character descriptions.

---

> ### Author Response · Authors · 2025-11-21
> **To reviewer 8oMt (4/5)**
>
> We also present additional statistical analyses for Section 4.2 below.
>
> ### Additional Analysis on Dynamic Reasoning Chains
> >Breaking down the analyses of reasoning chain strategies by the static and dynamic cases to see if these strategies are favored in the dynamic settings and connecting these back to RQ3 could strengthen the results.
>
> Thank you for suggesting that we need separate analysis for dynamic reasoning chains! In the initial submission of our paper, our analysis focuses on the static components (ambivalence, discomfort), so we conduct additional analysis on the dynamic part (value shift). By examining the reasoning chains, we unveil that overcommitment was particularly pronounced in the dynamic reasoning chains. We define overcommitment as excessive adherence to the previous perspective, hindering the model to adequately consider the current perspective. We design a prompt (Figure 30, added in the revision) to detect overcommitment and report the results in the 4th column of Figure 4(a) in our updated paper. All revisions are marked in blue to assist the reviewer in locating the changes. The updated Figure 4(a) shows that overcommitment appears more frequently in unsuccessful chains, contributing to the accuracy drop on the “current” question in dynamic settings, as observed in RQ3.
>
> Early commitment did not occur frequently enough in the reasoning chains to enable a meaningful analysis. We believe this is because the prompt explicitly instructs the model to reason from Character A’s current perspective (Figure 18), making it unlikely for the model to quickly resort to Character A’s previous perspective.
>
> ### Additional Statistical Results
> >the claims made in section 4.2 seem unjustified given the lack of statistical tests, error bars, and the very minimal difference between the values in the graphs for these claims.
>
> We validate our claim with statistical results. Using three models is not enough to validate our claim, so we run more experiments with other thinking models where we have access to the reasoning chains: Claude-Haiku-4.5 and Qwen3-14B. We present the Wilcoxon test result of each claim we make as follows:
>
> | Claim | p-value |
> |---------|----------|
> | Successful value-laden reasoning exhibits less backward chaining. |3.13e-2|
> | Successful value-laden reasoning exhibits less verification. |3.13e-2|
> | Early commitment occurs more frequently in unsuccessful chains. (static) |6.84e-3|
> | Overcommitment occurs more frequently in unsuccessful chains. (static) |6.84e-3|
> | Overcommitment occurs more frequently in unsuccessful chains. (dynamic) |6.25e-2|
> | Successful chains demonstrate greater emphasis on pragmatic ethics. |6.25e-2|
> | Successful chains demonstrate greater emphasis on right-based ethics. |3.13e-2|

---

> ### Author Response · Authors · 2025-11-21
> **To reviewer 8oMt (5/5)**
>
> > The paper could better contextualize their contributions in light of these studies, especially more recent ones
>
> Yes, we will enrich our related work section with recent moral decision-making literature in our camera-ready version. We will first discuss early datasets such as ([1],[2]), which introduce the task of moral decision-making for LLMs. We will then present more recent datasets that compare the model outputs with the “moral” decision ([3]) or human choices ([4],[5]), or examine the behavior or preferences of LLMs without a clear decision to compare against ([6],[7]). Our method contributes to the domain of moral decision-making by enabling verifiable evaluation of LLMs even in situations where no clear-cut ground truth choice exists, and by analyzing subtle aspects of moral reasoning that, to the best of our knowledge, have not been studied before. We will look for more relevant works and include those in the related work section of the final camera-ready version of our paper. Please let us know if you have any specific works that you think we should include, and directions you suggest to contextualize our work!
>
> Thank you for your suggestions!
>
> **References**
>
> [1] Aligning AI With Shared Human Values, Hendrycks et al., ICLR, 2021
>
> [2] Can machines learn morality? the delphi experiment, Jiang et al., Nat Mach Intell, 2025
>
> [3] When Ethics and Payoffs Diverge: LLM Agents in Morally Charged Social Dilemmas, Backmann et al., Arxiv, 2025
>
> [4] Large language models show amplified cognitive biases in moral decision-making, Cheung et al., PNAS, 2025
>
> [5] Are Rules Meant to be Broken? Understanding Multilingual Moral Reasoning as a Computational Pipeline with UniMoral, Kumar et al., ACL, 2025
>
> [6] Exploring the Choice Behavior of Large Language Models, Wu et al., ACL, 2025
>
> [7] The Staircase of Ethics: Probing LLM Value Priorities through Multi-Step Induction to Complex Moral Dilemmas, Wu et al., EMNLP, 2025

---

### Author Response · Authors · 2025-12-03
**Summary to AC**

We are grateful to the reviewers and the AC for their careful evaluation of our submission. We present a summary showing the contributions of CLASH and how we addressed the reviewers’ concerns.
***
## 1. Benchmark Overview
We present CLASH, a new benchmark for evaluating the value-based decision-making abilities of language models. Models are presented with 1) dilemmas and 2) value-based character descriptions, and should answer questions that demonstrate an understanding of both.
***
## 2. Contributions
### Key Features
1. **High-stakes, human-written, long-form dilemmas** that go beyond everyday topics, ensure narrative quality, and introduce additional complexity.
2. **Carefully designed character descriptions** that embed relationships among values, reflecting the complexity of real human value systems.
### CLASH enables the first study of
1. **Subtleties (ambivalence, discomfort, and value shift)** involved in decision-making.
2. **Conditional steerability**, a concept we introduce to assess whether a model follows value X when it conflicts with value Y, representing a more realistic setting in decision making compared to prior steerability work which mainly focuses on adhering to individual values.
***
## 3. Key Findings
As presented in our abstract, our key findings are:
1. Even strong proprietary models, such as GPT-5 and Claude-4-Sonnet, struggle with detecting ambivalence, achieving only 24.06 and 51.01 accuracy.
2. Although LLMs reasonably predict discomfort, they do not adequately comprehend value shifts.
3. Cognitive behaviors effective in other domains do not transfer to value reasoning. Instead, new failure patterns such as early commitment and overcommitment emerge.
4. The conditional steerability of LLMs is significantly correlated with their value preferences.
5. LLMs are more steerable when reasoning from a third-party perspective, although certain values (e.g., safety) benefit uniquely from first-person framing.
***
## 4. Summary of Responses on Reviewer Comments
| Comment | Response Summary |
|:----|:--------|
|Risk of dataset contamination (8oMt)|While some *situations* may exist in training data, other elements such as character descriptions are new and not seen. Thus, familiarity with the situations does not translate to correct reasoning for CLASH. Our expanded contamination analysis confirms that the dataset remains sufficiently challenging in difficult settings, scoring below 50\%.|
|Potential bias from using GPT-4o in dataset generation (8oMt)|GPT-4o was used only for initial drafts; all elements were reviewed and adjusted by humans when necessary. Our value-mapping process combines a specialized entailment model with GPT-4o to avoid overreliance, and is also inspected and validated by human reviewers.|
|Need for analysis of **dynamic** reasoning chains (8oMt)|In addition to our existing analysis of static reasoning chains, we added analysis of **dynamic** reasoning chains, showing overcommitment as a key failure mode. We conducted statistical analyses that support claims regarding the characteristics of successful reasoning chains in Section 4.2.|
|Why use the reasoning -> answer format? (Ehbp)|We note that reversing the order can bias the reasoning toward justifying a predetermined answer.|
|Careful design needed for first-person prompts (ViBw)| We have already designed the first-person prompt carefully to mirror third-person prompts with minimal lexical differences and matched token lengths.|
|Base preference may differ significantly in the first-person setting (ViBw)|Base preference for steerability is computed exclusively using the **base prompt**, which is identical in both first- and third-person settings.|
|Possible lack of control if character-description categories differ in length or explicitness of values (ViBw)|Differences in length across categories are an intended consequence of category design, as some categories are inherently more complex. We also provided examples showing that the explicitness of values is comparable across categories.|
|Cutoff of 16 pairs can skew results (ViBw)|We agree, and present results including all valid pairs. Although a few results lose statistical significance, the overall findings remain consistent.|
|Ethical considerations should be discussed (Ehbp, ycte)|We added an **Ethics Statement** addressing concerns related to high-stakes content and responsible use of models in sensitive domains.|
***
## 5. Conclusion for AC
CLASH is, to the best of our knowledge, the first dataset systematically examining decision complexities such as ambivalence, discomfort, and value shifts in high-stakes contexts with various perspectives. Through a comprehensive evaluation of 14 non-thinking and thinking models, followed by detailed analyses of reasoning chains and conditional steerability, we reveal key limitations and trends in complex value reasoning. CLASH offers a meaningful contribution to research on value understanding in language models.

---

### Meta-Review · Area_Chair_NHfd · 2026-01-06

**Summary:**

Major concerns included issues with dataset generation (e.g. reliance on GPT-4o), potential data contamination, lack of discussion of ethical considerations, focus solely on static reasoning, lack of statistical rigor, and evaluation on small sample sizes.

**Reviewer Concerns:**

Several reviewers already gave positive scores to begin with, and the authors made a substantial effort to address the concerns that reviewers did have. The authors have largely addressed some of the most major concerns regarding data contamination, ethical considerations, focus on static reasoning, evaluation on small sample sizes, and bias in dataset generation. The authors made a solid attempt to improve statistical rigor by adding several significance tests (some of which showed that some claims made by the authors may not be supported by the data), but they may not have done enough to satisfy the reviewers (e.g. adding error bars, CIs, multiple comparison correction, etc.).

**Reviewer Scores:**

I believe the scores of 6 and 8 would remain unchanged, but that it's feasible that one of the scores of 4 may have been increased to a 6 but not both.

---

### Decision · Program_Chairs · 2026-01-26

Accept (Poster)